# A Split-Marker System for CRISPR-Cas9 Genome Editing in Methylotrophic Yeasts

**DOI:** 10.3390/ijms24098173

**Published:** 2023-05-03

**Authors:** Azamat V. Karginov, Marina G. Tarutina, Anastasia R. Lapteva, Maria D. Pakhomova, Artur A. Galliamov, Sergey Y. Filkin, Alexey N. Fedorov, Michael O. Agaphonov

**Affiliations:** 1The Federal Research Center “Fundamentals of Biotechnology” of the Russian Academy of Sciences, Bach Institute of Biochemistry, 119071 Moscow, Russia; karginovaz@mail.ru (A.V.K.); mari.pakhomova.99@mail.ru (M.D.P.); arturens96@gmail.com (A.A.G.); s.filkin@fbras.ru (S.Y.F.); a.fedorov@fbras.ru (A.N.F.); 2National Research Center Kurchatov Institute, 123182 Moscow, Russia; m_tarutina@mail.ru (M.G.T.); anastasya.lapteva21@yandex.ru (A.R.L.); 3Kurchatov Genomic Center, NRC «Kurchatov Institute», 123182 Moscow, Russia

**Keywords:** Cas9, CRISPR, genome engineering, genome editing, methylotrophic yeast, *Ogataea*, *Komagataella*

## Abstract

Methylotrophic yeasts such as *Ogataea polymorpha* and *Komagataella phaffii* (sin. *Hansenula polymorpha* and *Pichia pastoris*, respectively) are commonly used in basic research and biotechnological applications, frequently those requiring genome modifications. However, the CRISPR-Cas9 genome editing approaches reported for these species so far are relatively complex and laborious. In this work we present an improved plasmid vector set for CRISPR-Cas9 genome editing in methylotrophic yeasts. This includes a plasmid encoding Cas9 with a nuclear localization signal and plasmids with a scaffold for the single guide RNA (sgRNA). Construction of a sgRNA gene for a particular target sequence requires only the insertion of a 24 bp oligonucleotide duplex into the scaffold. Prior to yeast transformation, each plasmid is cleaved at two sites, one of which is located within the selectable marker, so that the functional marker can be restored only via recombination of the Cas9-containing fragment with the sgRNA gene-containing fragment. This recombination leads to the formation of an autonomously replicating plasmid, which can be lost from yeast clones after acquisition of the required genome modification. The vector set allows the use of G418-resistance and *LEU2* auxotrophic selectable markers. The functionality of this setup has been demonstrated in *O. polymorpha*, *O. parapolymorpha*, *O. haglerorum* and *Komagataella phaffii*.

## 1. Introduction

The discovery of prokaryotic adaptive defense systems, which consist of the clustered regularly interspaced short palindromic repeat (CRISPR) and CRISPR-associated (*cas*) genes [1,2,3], has led to the development of efficient tools for genome editing in different organisms. Type II CRISPR-Cas systems include the Cas9 protein, which forms an endonuclease complex with two RNA molecules, tacrRNA and crRNA. The latter directs the endonuclease to the specific DNA sequence via complementary interaction to create a double strand break at the target site [4,5]. These two RNAs can be combined in a single guide RNA (sgRNA), which provides the Cas9 endonuclease complex with the same activity and specificity [5].

Introduction of CRISPR-Cas9 based genome editing revolutionized the availability of higher eukaryotes for molecular genetic manipulation. It facilitates the integration of foreign genes in a target genome locus, in addition to gene knockouts and replacements. This allows the development of transgenic plants [6] and animals [7,8] for industrial and research purposes. For example, the genome editing of bread wheat led to obtaining heritable resistance to powdery mildew [9]. Its use in *Arabidopsis* facilitated a study of transcriptional control of salt tolerance [10]. CRISPR-Cas9 based approaches are already in use the therapy of β-thalassemia [11] and transthyretin amyloidosis [12,13].

Although a wide spectrum of genome manipulation techniques has been developed for the *Saccharomyces cerevisiae* yeast, the CRISPR-Cas9 system provides significant advantages in a number of applications even in laboratory strains of this organism [14]. At the same time many traditional genome manipulation techniques are impractical in industrial *S. cerevisiae* strains and are not easy to use in a number of non-conventional yeast species, such as those which are extensively used in basic research and biotechnology. The main difficulties (namely host polyploidy, shortage of available and convenient selectable markers, and high frequency of non-homologous recombination) can be overcome with the use of the CRISPR-Cas9 system.

To ensure functionality of Cas9 in eukaryotic cells, the enzyme is commonly appended with a nuclear localization signal. While any sufficiently strong RNA polymerase II promoter can direct the expression of the Cas9-coding gene in eukaryotes, the choice of promotor for sgRNA is not always obvious. Indeed, RNA polymerase II transcribed RNAs are 5′-capped and 3′-polyadenylated, which may affect the sgRNA’s function, stability, and localization. This can be overcome by flanking the sgRNA with ribozymes, which then autocatalytically remove themselves from the sgRNA [15]. Alternatively, RNA polymerase I- or III-dependent promoters can be used, though such an approach is not always efficient [16]. A quite high efficiency of *S. cerevisiae* genome editing is achieved when the sequence coding for sgRNA with the hepatitis delta virus (HDV) ribozyme at the 5′ end and a polymerase III terminator at the 3′ end was fused to a tRNA gene by replacing is native terminator [17]. In this case, the ribozyme cleaves off the tRNA sequence, but does not remove itself from the mature sgRNA molecule. Importantly, this did not prevent proper targeting of the Cas9 endonuclease.

The thermotolerant methylotrophic yeasts *Ogataea polymorpha* and *O. parapolymorpha* (formerly *Hansenula polymorpha*) are extensively used as hosts for heterologous protein production, as well as being model organisms for studies of methanol utilization, peroxisome biogenesis and function [18], protein secretion and glycosylation [19], nitrite and nitrate assimilation [20], sugar metabolism [21], heat tolerance [22], and cell division [23]. Due to their ability to grow at elevated temperatures and utilize a broad range of substrates, these yeast species are also used as the basis for metabolically engineered strains that can produce valuable compounds [24,25,26]. All these features make the development of CRISPR-Cas9 tools for these yeasts highly valuable. At the same time, the approaches published so far appear to be laborious and complicated in their use. Numamoto et al. [27] have achieved a high frequency of genome editing with a single plasmid possessing both Cas9 and sgRNA genes. Such a plasmid was constructed in two steps. First, the sgRNA gene for each target was constructed using the fusion PCR technique, which was then inserted into a shuttle vector with a Cas9-encoding gene. This shuttle vector contained an *O. polymorpha URA3* selectable marker and a *S. cerevisiae* centromere sequence, however there was no indication whether this vector was able to autonomously replicate in *O. polymorpha*. Two reports [28,29] describe the use of CRISPR-Cas9 genome editing with a genome integrated Cas9-encoding gene. This is not convenient if the final strain should not possess unnecessary genes including a Cas9-encoding one.

Herein, we describe an easy-to-use CRISPR-Cas9 system which we developed for the genome editing of *Ogataea* yeast, utilizing an autonomously replicating vector obtained using the in vivo recombination of plasmid fragments possessing sgRNA and Cas9-encoding genes. A specific sgRNA gene was generated on a plasmid bearing the sgRNA gene scaffold interrupted with a fragment that was flanked by *Bsa*I restriction enzyme cleavage sites. This fragment is replaced with a 24 bp oligonucleotide duplex, providing the sgRNA with a targeting sequence. Since *Bsa*I is a type IIS restriction enzyme, the ligation with the oligonucleotide duplex can be performed in the presence of *Bsa*I (the so-called Golden Gate assembly [30]). This allows omitting the fragment purification step since the restriction enzyme is able to cut the original plasmid but not the resulting one. Moreover, the fragment being replaced carries a fluorescent protein gene, which allows selection of required *Escherichia coli* transformants using the absence of the specific fluorescence and/or color of colonies. Importantly, this genome editing system is effective in *O. polymorpha*, *O. parapolymorpha*, *O. haglerorum*, as well as in the more distantly related methylotrophic yeast *Komagataella phaffii*.

## 2. Results

### 2.1. Design of the Plasmids Bearing the Components of the CRISPR-Cas9 Genome Editing System

Previously [31], we successfully implemented a plasmid set developed by the Tom Ellis laboratory for *S. cerevisiae* [32] for inactivation of the *O. polymorpha MET8* gene. This set included shuttle vectors with a Cas9-encoding gene and sgRNA gene scaffold interrupted with a GFP-encoding gene, as well as the pWS082 plasmid, with only the GFP-interrupted sgRNA scaffold. Since the GFP-encoding gene was flanked by *Bsm*BI restriction sites, sgRNA gene for a specific target could be obtained with the replacement of the GFP-encoding gene with a 24-mer oligonucleotide duplex using the Golden Gate cloning procedure. This is performed in the pWS082 plasmid, and the resulting plasmid was cleaved at the sites flanking the sgRNA gene, while the Cas9 gene-containing vector was cleaved at *Bsm*BI sites within the sgRNA scaffold. This allowed the in vivo creation of an autonomously replicating plasmid bearing both genes via homologous recombination after the co-transformation of *S. cerevisiae* cells with the cleaved plasmids. However, this design was unlikely to work in *Ogataea* yeasts due to prevalence of non-homologous end joining (NHEJ) over homologous recombination. This was why we previously used a complete single plasmid from this set with the Cas9 gene, sgRNA, and the *LEU2* selectable marker for *O. polymorpha* genome editing [31]. Notably, the sgRNA gene in this system was represented by a tRNA gene whose terminator was replaced with HDV ribozyme fused to the guide RNA-encoding sequence, as in the study [17] mentioned in the Introduction.

However, the transformation efficiency was very low in *O. polymorpha*. The share of clones with the desired modification was also quite low. This was probably due to the inefficient autonomous replication of the plasmid in this yeast species and low expression of Cas9- and sgRNA-encoding genes. The latter could be due to a low efficiency of *S. cerevisiae* promoters in the heterologous host and due to codon optimization of the Cas9-encoding gene for *S. cerevisiae*. Moreover, only the plasmid with the *LEU2* selectable marker was suitable for the use in *O. polymorpha* among the plasmids available in this plasmid set since this marker was expressed at the sufficient level and ensures autonomous replication of the plasmid in this yeast.

To overcome these problems, we constructed two plasmids, pKAM944 and pKAM966, bearing a universal, dual-use yeast G418- and bacterial kanamycin-resistance markers (Figure 1A). The former plasmid encoded Cas9, while the latter possessed the scaffold for expression of a specific sgRNA. These plasmids shared some sequences for enabling *in vivo* recombination between specific plasmid fragments to combine Cas9- and sgRNA-encoding genes in one plasmid that could be episomally maintained in *Ogataea* cells. The sgRNA scaffold assembly in the pKAM966 plasmid followed the design reported for the pWS082 [17], but its expression was ensured by its fusion with the *O. polymorpha* TACtRNA-encoding gene (Figure 1A). In contrast to pWS082 this assembly was interrupted not with GFP-, but with an mCherry-encoding gene flanked by *Bsa*I sites, which also allowed the insertion of a specific targeting sequence via Golden Gate replacement of the fluorescent protein encoding gene with a 24-mer oligonucleotide duplex (Figure 1A,B). The *Escherichia coli* transformants possessing the plasmid with the required insert could be identified using their lack of mCherry fluorescence. Notably, we failed to obtain a plasmid possessing both a Cas9-encoding gene and sgRNA scaffold. We assume that simultaneous expression of these genes was toxic for *E. coli* cells. The pKAM966 possessed two 181 bp repeat sequences in distinct plasmid loci that could potentially cause a loss of the plasmid fragment due to recombination between these sequences. To prevent this, one of these repeats was removed and the resulting plasmid pKAM995 was also used in some experiments with the same efficacy as pKAM966.

### 2.2. Inactivation of the MET8 Gene in O. polymorpha

To test efficacy of this system in *O. polymorpha*, the *MET8* gene was chosen, since *Ogataea met8* mutants accumulate a fluorescent porphyrin compound that can be detected directly in growing colonies using red fluorescence exited with ~400 nm light [31]. For this purpose, the OpoMET8crU–OpoMET8crL oligonucleotide duplex (Table 1) was inserted into the pKAM995 plasmid, as described in Materials and Methods. The obtained pKAM995-based sgRNA-encoding plasmid was digested with *Sal*I and *Nru*I, cleaving the plasmids at the boundary of the Cas9-encoding gene fragment and within the G418/kanamycin resistance marker, respectively (Figure 1A). The pKAM944 plasmid was cleaved using *Pvu*I within the selectable marker and Cas9-encoding gene, or using *Sgf*I within the selectable marker and *Bcu*I outside the Cas9 gene. Unlike the former cleavage, the latter left the Cas9-encoding gene intact, but both of them created a fragment flanked by sequences, which overlapped with flanking sequences of *Sal*I–*Nru*I fragment of the sgRNA-encoding plasmid. This allowed for recombination between fragments of pKAM944 and the pKAM995-based plasmid that restored the functional selectable marker and created an autonomously replicating plasmid (Figure 1C). In case of the *Pvu*I cleavage, the functional Cas9 emerged only after the recombination, while the cleavage at *Sgf*I and *Bcu*I sites theoretically allowed Cas9 expression before the recombination.

The *O. polymorpha* 1B strain was transformed with a mix of the cleaved pKAM944 and the plasmid coding for the *OpoMET8*-targeting sgRNA, with or without a donor DNA fragment driving the gene deletion via homologous recombination in the Cas9-cleaved target locus. We expected that even in the absence of the donor DNA, gene inactivation could be achieved with the repair of the double strand break using erroneous non-homologous end joining (NHEJ) recombination. The standard transformation procedure using a G418 resistance marker is normally followed by a 1 h incubation of the cells in a non-selective medium prior to spreading them onto G418-containing plates. This allows the marker gene to be expressed before the cells are exposed to G148. In our case, prior to expression, the marker was reconstituted using the recombination of the transforming fragments. To allow enough time for this step, in the first experiment the cells were incubated in the non-selective medium for an increased amount of time (2 h). Some of the obtained transformants emitted red fluorescence upon irradiation with 405 nm light. All these clones were methionine auxotrophs. The transformation in the presence of the donor DNA fragment led to appearance of Met^−^ clones independently of the pKAM944 cleavage (Table 2, experiment 1). At the same time, without the donor DNA fragment, transformation with the *Pvu*I-cleaved plasmid did not produce any Met^−^ clones, while transformation with the *Sgf*I and *Bcu*I-cleaved plasmid produced such clones (Table 2, experiment 1).

We also tested whether the efficacy of the Cas9-directed mutagenesis depended on the duration of the incubation in the non-selective medium. After one hour’s incubation, no Met^−^ colonies were obtained, even in the presence of the donor DNA. Such clones appeared after increasing the incubation time to 2 h, while a 3 h incubation led to an additional increase in their number. The 3 h incubation allowed obtaining Met^−^ clones even with *Pvu*I-digested pKAM944 in the absence of the donor DNA fragment (Table 2, experiment 2). It is possible that placing cells onto a G418-containing medium somehow interferes with their survival after the Cas9-induced chromosomal double strand break, while incubation in non-selective conditions allows cells to complete the chromosomal break repair. Following this hypothesis, it is possible to infer that the double break repair by means of homologous recombination with the donor DNA may occur faster than NHEJ, and thus result in the increased number of Met^−^ clones.

### 2.3. Inactivation of the ADE2 Gene in O. haglerorum

To test the obtained plasmids in *O. haglerorum*, the *ADE2* gene was chosen as a target, since its inactivation leads to the accumulation of a red pigment and the *ade2* mutant colonies can then be identified by their red color. Based on pKAM966, two plasmids with gene coding for sgRNAs capable of targeting to different sites within the *O. haglerorum ADE2* gene were constructed. The obtained plasmids were digested as described in Figure 1 and used for transformation of *H. haglerorum* in combination with corresponding donor DNA and *Bcu*I–*Sgf*I-digested pKAM944. Such transformations, followed by incubation in a non-selective medium for 3 h, produced no colonies in most cases. This suggested that 3 h of incubation was insufficient to achieve a sufficient protection level for the marker expression. For this reason, the preincubation time was increased to 20 h. Since this led to a substantial increase in cell density, only a small portion of the cell suspension was spread onto the G418-containing plates. This led to obtaining a sufficient number of the transformants. The transformation with the sgRNA plasmid bearing the OhADE2F2–OhADE2R2 oligonucleotide duplex (Table 1) produced no red colonies, while 42% of transformants obtained with the sgRNA plasmid bearing OhADE2F1–OhADE2R1 oligonucleotides were red (Figure 2). PCR analysis of 79 of them revealed that 25% of them possessed *ADE2* deletion, which could have arisen due to the recombination with the donor DNA fragment. One clone possessed 736 bp deletion within the *ADE2* locus, which emerged independently of the donor DNA. The other clones experienced imperfect repair in the Cas9-generated double strand break via NHEJ, since sequencing of the target locus in seven of them revealed one-nucleotide deletion (in three clones) or insertion (in four clones) at the predicted Cas9 cleavage site (Figure 3).

### 2.4. The Use of the LEU2 Selectable Marker to Transform O. polymorpha and O. parapolymorpha with sgRNA and Cas9-Encoding Plasmids

Inactivation of some genes may increase sensitivity to G418 and other antibiotics. Specifically, some defects of protein glycosylation in the secretory pathway were shown to have such an effect in *O. polymorpha* and *O. parapolymorpha* [33,34,35,36]. Thus, the use of antibiotic resistance markers for the Cas9-mediated inactivation of such genes is hampered due to counter selection against the desired mutants. To overcome this problem, we constructed the pKAM977 plasmid with an sgRNA blank construct and a modified *S. cerevisiae LEU2* as a selectable marker. Cleavage of this plasmid with *Sal*I and *Eco91*I generated a fragment, which possessed only a portion of the *LEU2* marker and whose recombination with *Bst*DSI (*Dsa*I)-cleaved pKAM944 restored function to the *LEU2* marker and created a plasmid bearing both Cas9- and sgRNA-encoding genes (Figure 4). Notably, *Bst*DSI cleaves pKAM944 within the *LEU2* marker and at the first codon of the Cas9 open reading frame (ORF). This might delay Cas9 expression until completion of recombination of the transforming DNA fragments. To overcome this, the *LEU2* marker in pKAM944 was replaced with its fragment, ending with a newly generated *Hpa*I site (Appendix A). The cleavage of the resulting plasmid (designated pKAM1005) with *Hpa*I and *Bcu*I created a fragment which included the complete Cas9-encoding gene, and whose recombination with the sgRNA possessing fragment created a plasmid with a functional *LEU2* marker (Figure 4).

To test these plasmids in *O. polymorpha* and *O. parapolymorpha,* the *PMT1* gene was chosen, since its inactivation was known to increase G418 sensitivity. A 20 bp sequence (followed by a PAM motif revealed within both *OpoPMT1* and *OpaPMT1* ORFs) was chosen as a target for Cas9 endonuclease cleavage (Table 1, oligonucleotides PMT1_786U and PMT1_786L). Based on pKAM977, the plasmid coding for the sgRNA possessing this insert was constructed. A fragment of *OpaPMT1* locus with a deletion within its ORF was used as a donor DNA in both *O. polymorpha* and *O. parapolymorpha*.

The *O. parapolymorpha* DL1-L strain was transformed with the cleaved pKAM944 and pKAM1005 plasmids and combined with the sgRNA encoding plasmid, with or without the corresponding donor DNA fragment. Since inactivation of *PMT1* in *O. parapolymorpha* causes increased sensitivity to sodium dodecyl sulfate (SDS) and high temperature of incubation [35], the obtained transformants were tested for having these phenotypes. Indeed, some clones manifested both phenotypes independently of whether they were obtained using pKAM944 or pKAM1005, with or without the donor DNA fragment. PCR analysis of clones obtained with the donor DNA revealed that approximately half of them (six out of nine in case of pKAM944, and five out of eight in case of pKAM1005) possessed the deletion within *PMT1* due to its recombination with the donor DNA fragment. Thus, using the pKAM1005 plasmid while allowing Cas9-encoding gene expression before the restoration of the selectable marker did not improve the efficacy of the genome editing compared to pKAM944, whose Cas9-encoding gene was restored at the same moment as that of the selectable marker.

Notably, most of the *O. parapolymorpha* transformants selected for the *pmt1* mutant phenotypes had a stable Leu^+^ phenotype, indicating the integration of the transforming DNA. Possibly, this could be due to counter selection against those cells possessing a high copy number of the autonomous plasmid with Cas9- and sgRNA-encoding genes.

The pKAM944 plasmid contained the 2μ DNA fragment, which improved partitioning of episomal plasmids during cell division in *O. polymorpha* [37], but not in *O. parapolymorpha* (our unpublished observation). This centromere-like property decreased the plasmid copy number per cell and improved plasmid mitotic stability. We expected that this might have alleviated selective pressure against the episomal state of the sgRNA and Cas9-encoding plasmid. To test this, the *O. polymorpha* 1B strain was transformed with cleaved pKAM944, the sgRNA-encoding plasmid, and the donor DNA fragment. Among 112 transformants tested, 24 manifested increased SDS- and high temperature sensitivities. PCR analysis of these clones revealed that seven of them contained a *PMT1* deletion allele formed via its recombination with the donor DNA. Only one of them was unable to lose the Leu^+^ phenotype during growth in non-selective conditions.

Thus, the plasmids with *LEU2* selectable marker can be used in both *O. polymorpha* and *O. parapolymorpha*, however the efficacy of this approach was higher in the former species due to the much lower frequency of genome integration in the transforming DNA fragments carrying sgRNA- and Cas9-encoding genes.

### 2.5. Inactivation of the MET8 Gene in K. phaffii

Since *O. polymorpha*, *O. parapolymorpha* and *O. haglerorum* are very closely related species, it was not surprising that the obtained plasmid set was effective in all of them. We suggested that they might also be applicable to the more distant methylotrophic yeast *K. phaffii*. To test this, the *MET8* gene was chosen as a target, since this would allow us to test whether *MET8* inactivation in *K. phaffii* could lead to fluorescent porphyrin accumulation, as it does in *Ogataea* yeasts. For this purpose, a duplex of the oligonucleotides KpMET8F and KpMET8R was inserted into the pKAM966. This created a gene coding for sgRNA targeting Cas9 cleavage within the *KpMET8*. The obtained plasmid was digested with *Nru*I and *Sal*I and used for co-transformation of the *K. phaffii* X-33 strain with *Sgf*I–*Bcu*I-cleaved pKAM944. In this case, we did not include a donor DNA fragment and relied on obtaining mutations resulting from NHEJ. Approximately 10% (40 of 359) of the obtained transformants emitted red fluorescence upon irradiation with 405 nm light (Figure 5). All of these clones were methionine auxotrophs. This indicated inactivation of the *MET8* gene. Since the sgRNA- and Cas9-encoding plasmid (which we expected to be assembled in the transformed cells via in vivo recombination) possessed the autonomously replicating sequence *HARS6* originating from *O. parapolymorpha*, it was unclear whether this plasmid could be maintained autonomously in *Komagataella* yeast. To test this, the obtained Met^−^ clones were streaked on nonselective plates to obtain single colonies, which were then tested for G418 resistance. In most cases, only a portion of the subclones retained G418 resistance, confirming the capability of this plasmid for autonomous replication.

## 3. Discussion

Herein, we report the construction of a plasmid set, which facilitated CRISPR-Cas9 genome editing in methylotrophic yeasts, such as *O. polymorpha*, *O. parapolymorpha*, *O. haglerorum*, and *Komagataella phaffii*. Similar to the setup previously described for *S. cerevisiae* [17], the gene coding for the sgRNA with a specific targeting sequence was constructed with the insertion of a 24-mer oligonucleotide duplex into a ‘blank’ construct. Before the yeast transformation, the sgRNA and Cas9 encoding genes were presented separately on two different plasmids, whose restriction fragments were then able to form an autonomously replicating plasmid carrying both these genes via a process of homologous recombination once inside the yeast cell. To ensure correct plasmid assembly, the fragments participating in the recombination possess only a portion of a selectable marker, which is restored when the sgRNA encoding fragment is fused with the fragment of Cas9 encoding plasmid via homologous recombination in vivo. Such design implies that during the transformation, two (in absence of donor DNA) or three (with donor DNA) transforming DNA fragments should be simultaneously introduced into a yeast cell. This may appear to decrease transformation efficiency compared to the setup using a single circular plasmid possessing both sgRNA and Cas9-encoding genes; at the same time, such plasmids are very large (e.g., 13.5 kb [27]), which is not favorable for yeast transformation efficiency, while sharing these components between two different fragments may improve it. In addition, according to our previous experience, linearized DNA is more efficient in the transformation of at least *Ogataea* yeasts (our unpublished data), thus also favoring use of the ‘split’ setup. Unfortunately, we were unable to compare these setups since we failed to combine sgRNA and Cas9 encoding genes in one plasmid. A likely explanation for this is that simultaneous presence of these genes can be toxic to *Escherichia coli* cells. This may potentially be overcome by introducing an intron into the Cas9 ORF, as performed for the Cre recombinase gene [38,39,40,41]; however, this can be the subject of future study. Nevertheless, we successfully obtained transformants with a ‘split’ setup and performed genome editing in different methylotrophic yeasts. In addition, this setup should allow the use of multiple sgRNAs, similar to that described in *S. cerevisiae* [17], although this was not tested in practice.

## 4. Materials and Methods

### 4.1. Genetic Nomenclature Yeast Strains and Culture Conditions

The standard yeast nomenclature was used to designate genes of different yeast species. When necessary, the gene names were precluded with *Opo*, *Opa*, *Oh*, or *Kp* to indicate their origin from *O. polymorpha*, *O. parapolymorpha*, *O. haglerorum*, or *Komagataella phaffii*, respectively. The *O. polymorpha* 1B (*leu2 ade2*) [42], *O. parapolymorpha* DL1-L (*leu2*) [43], *O. haglerorum* VKPM Y-2584 [44], and *K. phaffii* (*Pichia pastoris*) X-33 (Invitrogen, Waltham, MA, USA) strains were used in this study.

### 4.2. Plasmids, Oligonucleotides and PCR Conditions

To avoid too complex a description of multiple steps involved in plasmid construction, in some cases only the composition of the plasmids obtained in this work will be explained in this section, while their maps and complete sequences are presented as Appendix A. The oligonucleotides used for plasmid construction and PCR analyses are listed in the Table 1.

The pKAM944 plasmid was constructed in several steps by the insertion of several elements into a pKAM544A vector possessing a selectable marker, which provided kanamycin resistance in *E. coli* and G418 resistance in *O. polymorpha*, *O. parapolymorpha*, *Komagataella phaffii*, and *S. cerevisiae* [45]. These elements were as follows: (i) *Komagataella phaffii LAT1* promoter-driven gene coding for Cas9 with SV40 NLS [46]; (ii) the *S. cerevisiae LEU2* gene with modified promoter for better performance in *Ogataea* cells [40]; (iii) *HARS6* [45] and the fragment of *S. cerevisiae* 2μ DNA providing an autonomously replicating plasmid with higher stability in *O. polymorpha* [37] (Figure 1 and Appendix A).

The pKAM966 plasmid was constructed from the pKAM944 plasmid by means of deletion within *Sal*I–*Sal*I fragment, removing the 3′ portion of the Cas9-encoding gene, and with the replacement of the *LEU2* gene in the resulting plasmid with the blank sgRNA gene construct. This construct consisted of a *O. polymorpha* TACtRNA-encoding gene whose terminator sequence was replaced with the HDV ribozyme, followed by the *Bsa*I-flanked gene coding for mCherry under the control of an *E. coli* promoter, and sgRNA scaffold with the *S. cerevisiae SNR52* terminator sequence (Figure 1 and Appendix A). This plasmid possessed two 181 bp repeat regions. This did not noticeably affect its use in most experiments. Nevertheless, one of the repeat sequences was removed by deleting the *Pci*I–*Mlu*I fragment to obtain the pKAM995 plasmid (Appendix A), which was also used in some experiments with the same efficacy as pKAM966.

The pKAM977 plasmid (Appendix A) was constructed with the replacement of the 717 bp *Apa*LI–*Mlu*I fragment in pKAM966 with the 1733 bp *Apa*LI–*Xho*I fragment of pKAM944 bearing the *LEU2* marker. To allow ligation, the *Xho*I- and *Mlu*I-generated overhangs were filled in with the Klenow enzyme.

The pKAM1005 plasmid (Appendix A) was constructed with the replacement of the 2088 bp *Bcu*I–*Eco*91I fragment of pKAM944 with a 959 bp fragment of the *Eco*91I-cleaved PCR product that we obtained with primers oriA and ScLEU2_989HpaI (Table 1), using pKAM944 as a template and high fidelity PfuSE polymerase (SibEnzyme, Novosibirsk, Russia). To allow ligation, the *Bcu*I-generated overhang was filled in by the Klenow enzyme.

To obtain sgRNA genes for specific targets, the mCherry gene in the plasmids pKAM966, pKAM995, or pKAM977 was replaced with a specific 24-mer oligonucleotide duplex (Table 1). To achieve this, the plasmid aliquot was incubated with *Bsa*I in the recommended buffer at 37 °C for two hours. Then, ATP solution (final concentration 0.5 mM), oligonucleotide duplex (final concentration 0.3 μM) and T4 DNA ligase were added and the mix was incubated overnight at 37 °C to be used for *E. coli* transformation. The transformants lacking mCherry fluorescence possessed an insertion of the oligonucleotide duplex.

Construction of the pAM913 plasmid, which was the source of donor DNA for *OpoMET8* deletion (Appendix A), has been described previously [31].

To obtain donor DNA for *O. parapolymorpha PMT1* inactivation, *Sal*I–*Xba*I deletion was achieved in the plasmid p90-203 possessing this gene locus [35]. Then, the *Nru*I–*Bam*HI 1298 bp fragment was deleted to obtain the pDA2 plasmid. The donor DNA fragment was obtained with the cleavage of the pDA2 plasmid using *Sal*I and *Eco*RV.

The donor DNA for the *ADE2* gene inactivation in *O. haglerorum* (Appendix A) was obtained with the PCR amplification of the recombination arms, using primer pair Up-ADE2-F and Up-ADE2-R for the upstream arm, and Dn-ADE2-F Dn-ADE2-R for the downstream arm (Table 1). Then, these PCR products were fused using PCR with the primers Up-ADE2-F and Dn-ADE2-R. The high fidelity PfuSE polymerase (SibEnzyme, Novosibirsk, Russia) was used to obtain these products. The resulting PCR product represented the *OhADE2* locus with 1648 bp deletion within the coding region of this gene. The recombination arms have the same length of 553 bp.

The regular Taq polymerase (Evrogen, Moscow, Russia) was used for the PCR analysis of the obtained transformants.

### 4.3. Yeast Transformation

*O. polymorpha* and *O. parapolymorpha* were transformed using the Li-acetate method [47], with some modifications [31]. *O. haglerorum* and *K. phaffii* were transformed with electroporation. The *O. haglerorum* competent cells were prepared according to the protocol described in Saraya et al. [48]. 60 μL of defrosted cell suspension was mixed with transforming DNA and electroporated in a 2 mm electroporation cuvette using the GenePulser Xcell (Bio-Rad) under following parameters: 1.5 kV (7.5 kV/cm), 200 Ω, 25 μF. *K. phaffii* competent cells were prepared according to protocol described in Lin-Cereghino et al. [49]. 40 μL of competent cells were mixed with transforming DNA and electroporated in a 2 mm cuvette using a Micropulser (Bio-Rad, Hercules, CA, USA) with the following settings: 2kV (10 kV/cm), 200 Ω, and 25 μF.

Prior to yeast transformation the plasmids were digested with restriction enzymes indicated in Results. The obtained fragments were precipitated with isopropanol, washed twice with 70% ethanol, dissolved in water, and directly used for transformation without isolation of individual fragments. The PCR product used as a donor DNA for *OhADE2* deletion was purified using the Cleanup Mini kit (Evrogen, Moscow, Russia).

## Figures and Tables

**Figure 1 ijms-24-08173-f001:**
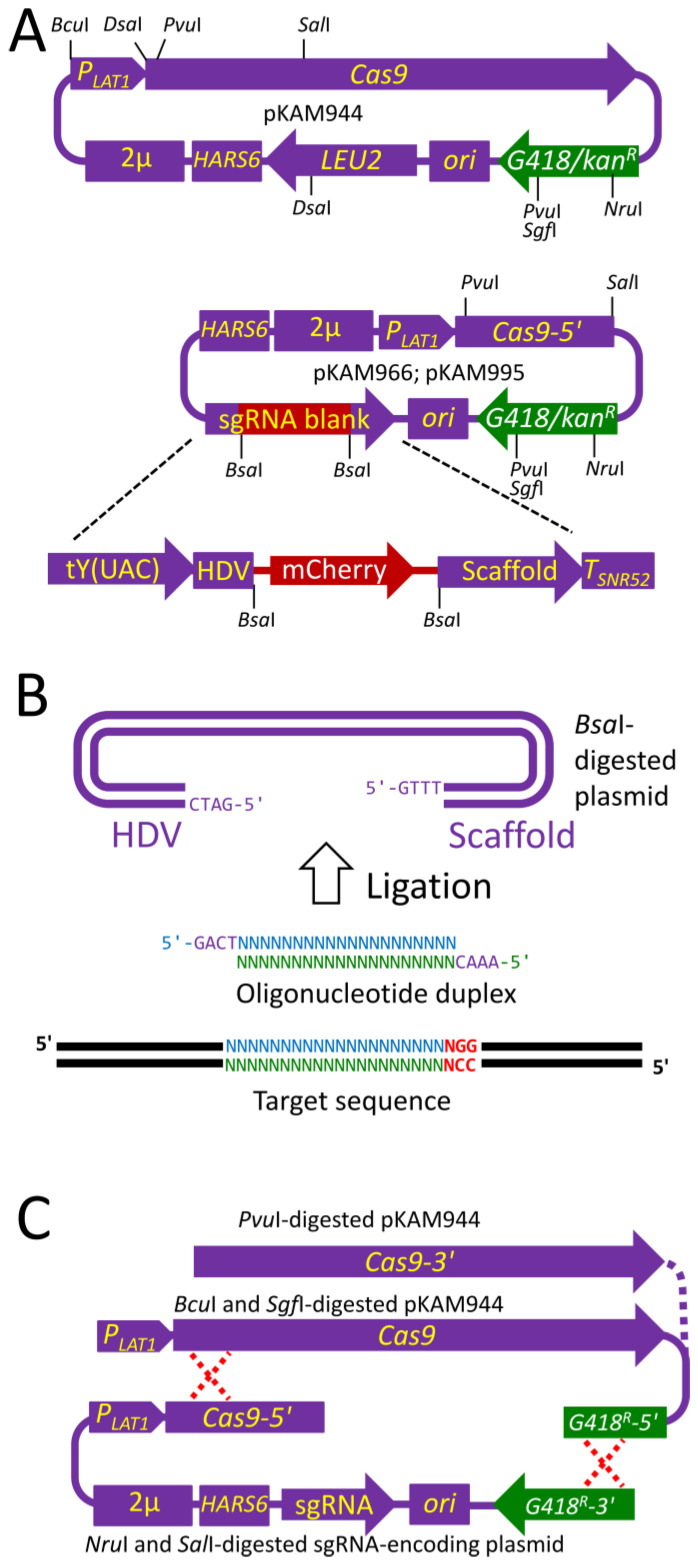
Genetic constructs used to introduce Cas9- and sgRNA-encoding genes into yeast cells. (**A**) Schemes of pKAM944 and pKAM966 plasmids. *P_LAT1_*, *Komagataella phaffii LAT1* promoter; *Cas9*, gene coding for Cas9 with SV40 NLS; *ori*, the bacterial ColE1 replication origin; 2μ, fragment of *S. cerevisiae* 2μ DNA bearing STB locus; *HARS6*, an *O. parapolymorpha* autonomously replicating sequence; *G418/kan^R^*, a selectable marker providing G418 resistance in yeast and kanamycin resistance in *E. coli*; tY(UAC), *O. polymorpha* gene encoding tyrosine tRNA without the terminator sequence, mCherry, a gene coding for mCherry under control of the *E. coli* promoter; Scaffold, the sgRNA-encoding sequence lacking the targeting part; *T_SNR52_*, a *S. cerevisiae SNR52* terminator. (**B**) Scheme of insertion of the targeting sequence into the plasmid with the blank sgRNA construct. The upper and lower strands of the target sequence are shown as blue and green N characters, respectively. The PAM sequence is shown in red. (**C**) Scheme of the in vivo recombination (crossed dashed red lines) of plasmid fragments creating the autonomously replicating plasmid bearing Cas9 and sgRNA encoding genes.

**Figure 2 ijms-24-08173-f002:**
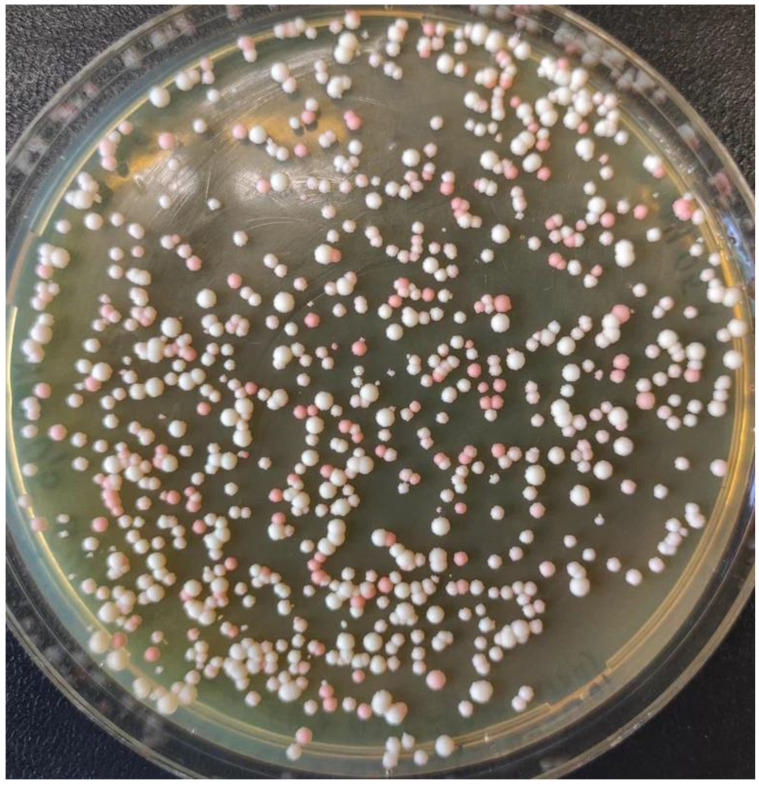
*O. haglerorum* transformants obtained using a mix of *Sgf*I–*Bcu*I-digested pKAM944, the donor DNA for *OhADE2* deletion, and the *Sal*I–*Nru*I-digested plasmid encoding the sgRNA with an inserted OhADE2F1–OhADE2R1 oligonucleotide duplex.

**Figure 3 ijms-24-08173-f003:**
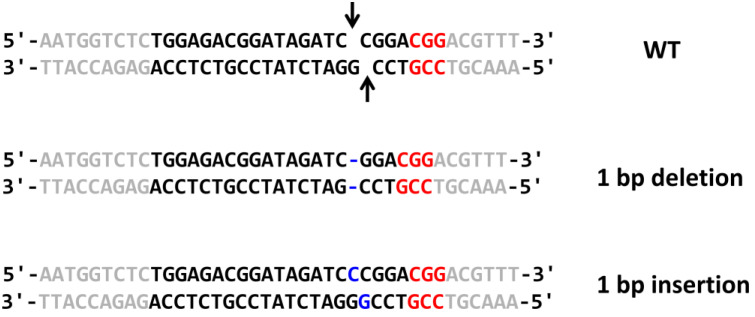
Sequences of the Cas9 target site in the *O. haglerorum* wild-type strain (WT) and Ade^−^ transformants possessing 1 bp deletion or insertion within this site. Sequence of the 20 bp target site and its flanking sequences are shown in black and gray, respectively; PAM sequence is shown in red; mutations are shown in blue; arrows indicate predicted Cas9 cleavage site.

**Figure 4 ijms-24-08173-f004:**
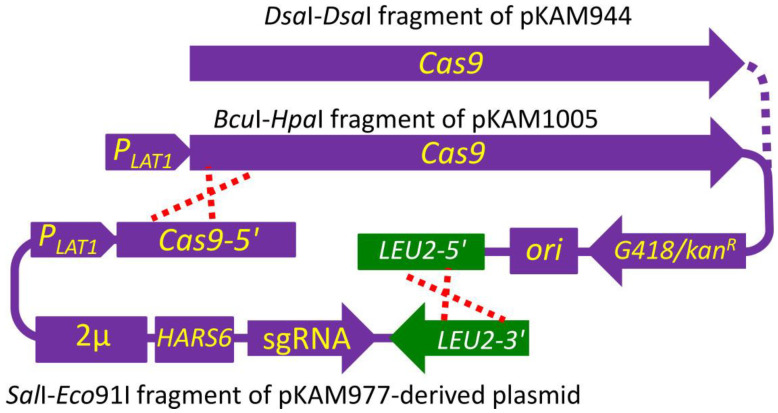
Scheme of the in vivo recombination (crossed dashed red lines) of plasmid fragments producing the autonomously replicating plasmid bearing Cas9- and sgRNA-encoding genes and a *LEU2* selectable marker. Designations as in the Figure 1.

**Figure 5 ijms-24-08173-f005:**
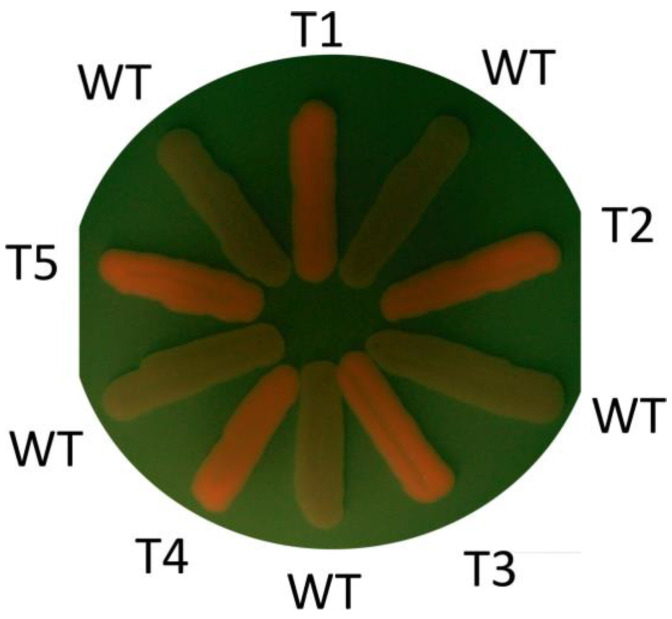
Porphyrin fluorescence in *K. phaffii met8* mutants obtained with the transformation of the X-33 strain with a mix of *Sgf*I–*Bcu*I-digested pKAM944 and the *Sal*I–*Nru*I-digested plasmid coding sgRNA, with the insertion of the KpMET8F–KpMET8R oligonucleotide duplex. Mutant clones (T1–T5) and untransformed strain (WT) were patched onto a YPD plate, incubated for two days at 30 °C and photographed through a yellow filter under illumination with 405 nm light emitting diode.

**Table 1 ijms-24-08173-t001:** Oligonucleotides used in this study.

Name	5′-3′ Sequence	Purpose
oriA	CCTATGGAAAAACGCCAGCAA	PCR for plasmid construction
ScLEU2_989HpaI	TTTGTTAACTGCATCTTCAATGGC	PCR for plasmid construction
OpoMET8crU	GACTAGCAGATGCACAGATCACAG	OpoMET8 sgRNA
OpoMET8crL	AAACCTGTGATCTGTGCATCTGCT	OpoMET8 sgRNA
MET8opoAU1	GTCCTTGGAGACACCTTACC	OpoMET8 PCR analysis
MET8opoAL1	CCGCTCGAAATGCGCTCTAT	OpoMET8 PCR analysis
KpMET8F	GATCTTGGGTCAGTAGTAAGACAG	KpMET8 sgRNA
KpMET8R	AAACCTGTCTTACTACTGACCCAA	KpMET8 sgRNA
OhADE2F1	GACTTGGAGACGGATAGATCCGGA	OhADE2 sgRNA
OhADE2R1	AAACTCCGGATCTATCCGTCTCCA	OhADE2 sgRNA
OhADE2F2	GACTAGCAGTGCTCTCAACAGCGG	OhADE2 sgRNA
OhADE2R2	AAACCCGCTGTTGAGAGCACTGCT	OhADE2 sgRNA
Up-ADE2-F	GATGTCGAAGTCAACAAAATCCAGG	Construction of donor DNA for OhADE2 deletion
Up-ADE2-R	CCTACGACCTTCGAGTCCATGATAC	Construction of donor DNA for OhADE2 deletion
Dn-ADE2-F	TACATTAATTTAATTAGTATCATGGACTCGAAGGTCGTAGGGGCTCTGTTGGCTACGAGGAG	Construction of donor DNA for OhADE2 deletion
Dn-ADE2-R	CTCCATTTCCACCCTTTCCCGAC	Construction of donor DNA for OhADE2 deletion
Up-ADE2-sq-F	CCTTCTGTCGTCTACCGTTCTCTGTC	OhADE2 PCR and sequencing analyses
Dn-ADE2-sq-R	CGACAGCGGACACATAGACGTTGC	OhADE2 PCR and sequencing analyses
ADE2-4-F	GTATCATGGACTCGAAGGTCGTAGG	OhADE2 PCR and sequencing analyses
ADE2-847-R	CTCAAACTGGCTCGTAACACACGC	OhADE2 PCR and sequencing analyses
PMT1_786U	GACTTCCCACAACCATCTGTACGA	OpaPMT1 sgRNA
PMT1_786L	AAACTCGTACAGATGGTTGTGGGA	OpaPMT1 sgRNA

**Table 2 ijms-24-08173-t002:** Emergence of Met^−^ clones in transformation of *O. polymorpha* with a mix of cleaved pKAM944 plasmid and pKAM966-based plasmid with a OpoMET8crU-OpoMET8crL oligonucleotide duplex insert.

Experiment	pKAM944 Cut	Donor DNA	Preincubation (h)	# of Transformants	# of Met^−^	% of Met^−^
1	PvuI	+	2	110	22	20
1	PvuI	-	2	49	0	0
1	SgfI BcuI	+	2	244	29	12
1	SgfI BcuI	-	2	158	13	8
2	SgfI BcuI	+	1	38	0	0
2	SgfI BcuI	+	2	49	3	6
2	SgfI BcuI	+	3	43	6	14
2	SgfI BcuI	-	3	88	16	18
2	PvuI	-	3	81	11	14

## Data Availability

All data generated or analyzed during this study are included in this published article.

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
