# Peer review of "A Split-Marker System for CRISPR-Cas9 Genome Editing in Methylotrophic Yeasts"

_ijms, 2023, doi:10.3390/ijms24098173_

Round 1
Reviewer 1 Report
The manuscript by Karginov et al. described improved plasmid vector set for CRISPR-Cas9 genome editing in methylotrophic yeasts. Functionality of this setup has been demonstrated in O. polymorpha, O. parapolymorpha, O. haglerorum and Komagataella phaffii. The manuscript contains interesting applications of CRISPR-Cas9 genome editing system in different thermotolerant methylotrophic yeasts. However, I suggest that several issues need to be identified and revised, and my comments are detailed in attachment.

No comments.
Author Response
We have corrected all the indicated positions as well as some errors created by our citation manager, which we found ourselves. However, there were some exceptions: we have not added periods after the journal names Yeast and PLoS One, since these are not abbreviations.
We have included an additional paragraph into the Introduction section describing and citing some examples of the successful use of CRISPR-Cas9 systems in different organisms.
Reviewer 2 Report
The authors have demonstrated a common CRISPR-Cas9 based genome editing method for multiple yeasts. The uniqueness of this study is that these yeasts are not model yeasts like Saccharomyces and Pichia so are relatively difficult to engineer. Therefore, availability of improved CRISPR toolbox will help in using methylotrophic yeasts for biotechnological applications. Overall, the experimental design is accurate, the manuscript written well and results are easy to follow. I have a few comments for authors to address before publication:
a) Legends for Table 2 should provide more detail. Were there any other experiments performed to confirm Met deletion such as accumulation of fluorescent porphyrin? If yes, please include data.
b) In material and method add a molecular biology section, covering PCR condition, gel purification, fragment clean up. It is not clear after digestion, how many fragments were obtained and How many fragments were included for transformation and in vivo recombination in yeast. Because if all the fragments were included there are chances that the original plasmid can recombine. Arte these fragments were separated and purified. Include all these details for result replication.
c) Similar to plasmid map and sequences, also include donor sequences that were used in supplementary. It will make easier for reader to understand and data replication.
Author Response
a) Q: Legends for Table 2 should provide more detail. Were there any other experiments performed to confirm Met deletion such as accumulation of fluorescent porphyrin? If yes, please include data.
R: Analysis of porphyrin accumulation by Ogataea met8 mutants has been described previously by our group. Here we detected the porphyrin accumulation by specific fluorescence of colonies. All fluorescent transformants were methionine auxotrophs. We included an explanation into the text. This means that all the Met- clones indicated in Table 2 accumulated fluorescent porphyrins. We hope that the explanation we added to the text is sufficient and so decided that an extension of the table title is not required.
b) Q: In material and method add a molecular biology section, covering PCR condition, gel purification, fragment clean up. It is not clear after digestion, how many fragments were obtained and How many fragments were included for transformation and in vivo recombination in yeast. Because if all the fragments were included there are chances that the original plasmid can recombine. Arte these fragments were separated and purified. Include all these details for result replication.
R: We extended the existing sections of Materials and Methods to describe PCR procedures and preparation of plasmid fragments for yeast transformation. Actually, we did not separate the plasmid fragments, since we did not expect them to be capable of efficiently re-assembling into the original plasmid. This requires a precise assembly of two fragments via NHEJ recombination that we think is unlikely, as opposed to homologous recombination creating the sgRNA- Cas9- expressing plasmid. Actually, our data strongly support this expectation since we did not see much difference between plasmids with a complete and incomplete LEU2 marker. In addition the fragment purification step would make this approach more laborious and less attractive to potential users.
c) Q: Similar to plasmid map and sequences, also include donor sequences that were used in supplementary. It will make easier for reader to understand and data replication.
R: The sequences and schemes of the donor DNA fragments have been included into the supplement.
Author Response
Q1- The authors have used Golden Gate procedure to construct sgRNA. This procedure must be defined and described in 3-4 sentences. Also, it is necessary to mention the advantages/properties of this procedure rather than other methods.
R: An explanation of the Golden Gate assembly and its advantages has been included into the Introduction.
Q2- According to the result, the low transformation efficiency of O. polymorpha might be the result of codon optimization of Cas9-encoding gene for S. cerevisiae. Did the authors change the codon optimization to overcome this problem?
R: The codon usage was the reason why we decided to use the human optimized Cas9 instead of the gene with a S. cerevisiae codon usage. The former’s codon usage seems much more preferable for Ogataea yeast than the latter one. However we did not study whether this had a significant impact on the expression level, since this was not the main goal of our work. The expression level of the gene we finally used in our constructs was sufficient for successful genome editing.
Q3- In the discussion section the authors claim that “A likely explanation for this is that simultaneous presence of these genes can be toxic to Escherichia coli cells.” This must be explained in details. Why these genes are inducing toxicity to E.coli?
R: Actually we do not know whether the simultaneous presence of Cas9 and unfunctional sgRNA genes on a plasmid is toxic. We just suggested it as a possible explaination why we could not obtain such a plasmid. We were able to obtain E. coli transformants, which according to the PCR analysis and phenotype did carry such a plasmid, but these transformants grew very slowly on solid medium, were unable to grow in liquid medium, and their cells tended to lyse. Most importantly. we failed to isolate the plasmid from these cells. We did not describe this in the manuscript since this was not the topic of our research. However we decided to mention possible incompatibility of Cas9 and sgRNA genes in E.coli since our approach escapes this potential problem.
Q4- In the discussion section, the authors say: “ This may appear to decrease transformation efficiency compared to the setup using a single circular plasmid possessing both sgRNA and Cas9-encoding genes.” Considering this important point, can this method be used for industrial application? What are the major challenges to scale-up such method?
R: We are sorry, but we do not understand why yeast transformation should be scaled up for industrial applications.
Q5- Around 25% of the references are self-citation. This must be reduced to a normal range .
R: This relates only to one of the co-authors, who has been working with Ogataea yeast approximately 30 years and was involved in construction of many strains and plasmids, which were described in different papers. Some of observations and materials described in those publications were used in this manuscript and we have to cite them. To reduce the share of self-citation, we have added several more papers cited in the Introduction.
Q6- Overall, the paper has been prepared in an organized manner. However, the content of the manuscript is more suitable for a journal that publishes novel “protocols”
R: We disagree with this statement. Indeed, our work has led to development of a protocol but this required considerable research efforts, which are described in the manuscript. In addition, the paper is the first to describe the possibility of obtaining novel auxotrophic mutants via inactivation of the MET8 in Komagataella phaffii, and shows that this gene deletion causes accumulation of fluorescent porphyrins in in this yeast.